# Performance of MoS_2_/Zr Composite Coatings at Different Deposition Temperatures

**DOI:** 10.3390/ma14175100

**Published:** 2021-09-06

**Authors:** Wenlong Song, Kai Sun, Guangming Zhao, Long Zhu, Shoujun Wang, Tianya Li

**Affiliations:** 1Jining Mining Group-Haina Technology Electromechanical Co., Ltd., Jining 272100, China; 13675370099@139.com (K.S.); jnzgming@126.com (G.Z.); 13562793933@139.com (L.Z.); 2School of Industry, Jining University, Qufu 273155, China; shoujun0531@163.com (S.W.); lty123456202108@163.com (T.L.)

**Keywords:** deposition temperature, MoS_2_/Zr coatings, mechanical properties, tribological properties

## Abstract

The properties of the MoS_2_/Zr coatings can be significantly affected by the deposition temperature. In this study, the MoS_2_/Zr composite coatings were fabricated on the cemented carbide surface, utilizing the duplex deposition technology at various deposition temperatures. The effects of deposition temperature on the mechanical and friction properties of the MoS_2_/Zr coatings were systematically studied. Results exhibited that as the deposition temperature increased, the adhesion force increased first and then decreased, and the coating thickness and micro-hardness gradually increased. Dry sliding tests against a hardened steel ring showed that the tribological behaviors and wear mechanisms of the MoS_2_/Zr coatings varied with deposition temperature, which were due to the changing mechanical properties of coatings caused by the temperature. The coatings deposited at a temperature of 180 °C and 200 °C possessed preferable comprehensive mechanical and tribological properties.

## 1. Introduction

Surface coating technology is an efficient approach to enhance the friction and wear properties of substrate materials, and has been extensively applied in numerous engineering applications [1].

The first generation coatings featured TiC [2], TiN [3,4], ZrN [5] and Al_2_O_3_ [6,7] as the hard coatings, and were applied to improve the material durability in aggressive environments. The superior performances of the coated tools improve their application in manufacturing processes, such as drilling, turning, boring and as a protective layer on dies. The development of material science and coating technology has resulted in the development of more advanced coatings, e.g., AlCrN [8], TiCN [9], TiAlN [10,11], TiAlSiN [12,13,14], which further improved the industry applications of coatings, owing to a much higher surface hardness with excellent wear resistance.

Molybdenum disulphide (MoS_2_) is a widely applied solid lubricant with a hexagonal structure [15]. During sliding friction tests in the absence of oxygen and/or water vapor, MoS_2_ coating had good self-lubricating characteristics with a friction coefficient of less than 0.1 [16,17]. The mechanism of lubrication of MoS_2_ is due to the high adhesion to material surfaces and a low shear strength along the base plane of the hexagonal structure [18]. To enhance the tribological performance of the pure MoS_2_ coating, the MoS_2_/metal composite coatings, such as MoS_2_/Ti [19,20], MoS_2_/Cr [21,22], MoS_2_/W [23], MoS_2_/C [24], MoS_2_/Y [25] and MoS_2_/Zr [26,27], have been fabricated with more favorable mechanical properties and tribological properties. The MoS_2_/Zr composite coatings have been widely used in industrial productions, such as molds, bearing rings, cylinder blocks of engines and machining tools.

According to previous results, MoS_2_/Zr coatings are beneficial to improve the friction and wear properties of cemented carbide in friction tests [26,27] and dry cutting tests [28,29]. In addition, the influences of experimental conditions, such as applied force and speed, on the tribological properties have also been comprehensively researched [26,29,30]. However, the influence of deposition temperature on the performance of MoS_2_/Zr coatings and the mechanisms underlying this have not been reported in detail.

In this study, the duplex deposition technology, combining medium frequency magnetron sputtering technology (MFMS) with multi arc ion plating technology (MAIP) is utilized to deposit the composite coatings on the carbide substrate under different temperature conditions. The purpose of this research is to comprehensively study the effects of deposition temperature on the adhesive strength, thickness, microhardness and tribological performance of the coatings. The results can provide a useful reference for the further industrial application of MoS_2_/Zr composite coatings.

## 2. Materials and Methods

### 2.1. MoS_2_/Zr Coating Preparation

The MoS_2_/Zr composite coatings were deposited on a carbide sample (WC + 15%TiC + 6%Co) using the MFMS technology combined with the MAIP method. All the samples were first polished and cleaned ultrasonically, then were placed in an AS-585 multifunctional coating equipment (Dalian Vacuum Technology Co., Ltd., Dalian, China), which was equipped with two MoS_2_ targets (MFMS), a Zr target (MAIP) and a Ti target (MAIP). In order to enhance the adhesion strength of the coatings with the carbide substrate, a Ti layer was first sputtered for 5–6 min. The preparation conditions of MoS_2_/Zr coatings are shown in Table 1. The MoS_2_/Zr coatings deposited at different temperatures were named T50, T120, T150, T180, T200, T250 and T300, according to the deposition temperatures.

A MT-4000 material properties tester (Lanzhou Huahui Instrument Co., Ltd., Lanzhou, China) was applied to measure the adhesion strength and coating thickness, by scratching a diamond tip of a radius 200 μm on the coating surface. The scratch test for adhesion was performed utilizing a 0–100 N normal load, a 100 N/min load increasing rate and a constant 10 mm stroke. The scratch test for coating thickness was conducted using a 6 mm stroke and a 60 s test duration.

The surface hardness of the coatings was tested using a micro-hardness instrument (MH-6, Shanghai Testing Instrument Co., Ltd., Shanghai, China) at an applied load of 0.1 N to eliminate the effect of the carbide substrate.

### 2.2. Friction Tests

The tribological behaviors and mechanisms of the MoS_2_/Zr coatings were investigated utilizing an MRH-3 block-on-ring friction tester (Jinan Shijin Co., Ltd., Jinan, China). The schematic diagram of the friction test is indicated in Figure 1. The upper block (15 mm× 15 mm× 4.5 mm) was the MoS_2_/Zr coated sample, and the counterpart (50 mm× 35 mm× 15 mm) was an AISI 5140 quench-hardened steel ring with a surface hardness of 5.1 GPa. The specimens were cleaned with alcohol, and then cleaned ultrasonically with acetone. The coated block was fastened on a fixture, and the 5140 ring was rotated at a load of 10 N and a speed of 250 mm/s. The average friction coefficient was obtained by computing the ratio of the tangential load to normal load.

All the tests were performed three times and the average values were taken. To determine the friction and wear characteristics of the MoS_2_/Zr coated samples, investigations were also carried out with scanning electron microscope (SEM, INCA Penta FETXS, Oxford, UK) and energy dispersive X-ray spectrometer (EDX, D8 ADVANCE, Bruker, Germany).

## 3. Results and Discussion

### 3.1. Mechanical Properties

Figure 2 exhibits the surface and cross-section micrographs of the T180 sample. It can be seen that the coatings possessed fine and dense structures, which was probably attributed to the effect of Zr solution strengthening in the composite coatings [26,27,28]. In addition, the coatings had excellent adhesion with the substrate, and the coating thickness was about 2.2 μm. Figure 2c–f indicates the EDX line scanning of S, Mo, Zr and Ti elements along the cross section of the coatings in Figure 2b. As indicated in the figures, the element content of Mo, S and Zr increased significantly in the surface coatings, and these elements were relatively uniformly distributed in the direction of the coating thickness. At the same time, there existed one peak of Ti content at the coating-substrate interface (Figure 2f), which was consistent with the thin Ti transition layer between the coatings and substrate.

Figure 3 indicates the X-ray diffraction analysis result in the coating surface of the T180 sample. It can be seen that there was no obvious MoS_2_ diffraction peak or broad diffraction peak. This was explained in that the addition of Zr element could destabilize it. The test results also revealed that the addition of metals (Ti, Cr, Zr, etc.) into the MoS_2_ coating, to prepare the MoS_2_/metal coatings, can result in the amorphous microstructure of MoS_2_ [21,22,23,26,30].

The adhesion of coatings with carbide substrate can be evaluated according to the curve variation of the friction force and the friction coefficient. Figure 4 shows the curve of adhesion force in a scratch test for the T180 sample. It revealed that the value of the friction coefficient and curve slope of the friction force were both very small and relatively steady in the initial stage of the scratch process. As the scratch force reached about 68 N, the values of friction force and friction coefficient began to rise quickly, due to the wear and tear of coatings; while after 75 N, the coatings were completely worn out, and the friction curves achieved a steady state with relatively high values. Thus, the adhesion force for the T180 sample was at least 68 N.

The scratch micrographs and corresponding EDX analysis of the sliding track are indicated in Figure 5. It can be seen that there existed serious mechanical plows and delamination in the scratch area (Figure 5a). Meanwhile, the carbide substrate was already partly exposed, and a small amount of the coating elements could be detected in Figure 5c,d, which can lead to the increase of the friction coefficient and friction force. The analysis results were consistent with the test curves of friction force and friction coefficient in Figure 4. Figure 6 shows the average values of adhesion force. It can be seen that the adhesion force gradually increased and achieved the best value (68 N) at 180 °C, with an increase in deposition temperature from 50 °C to 180 °C. However, a further increase in deposition temperature caused a decrease in adhesion property, and the adhesion force went down to about 45 N at a temperature of 300 °C.

The coating thickness can be determined by measuring the difference in surface height between the coatings and uncoated substrate. Figure 7 indicates the thickness curve of the T180 sample in the scratch test. It can be seen that the coating thickness was determined as about 2.20 μm, which was in accordance with the value from the cross-section morphology of the coatings in Figure 2b. Figure 8 illustrates the average values of coating thickness at various deposition temperatures. It revealed that the thickness of coatings ran in a similar linear increase as deposition temperature rose. As deposition temperature increased from 50 °C to 300 °C, the thickness of the coatings increased from about 1.60 μm to 2.42 μm. This was probably because the increasing temperature was conducive to improving the sputtering energy, which can lead to the increase of the deposition rate and coating thickness [31].

Figure 9 illustrates the average values of the surface micro-hardness. It can be seen that the surface micro-hardness of the coatings significantly increased firstly with the deposition temperature, from 8.20 ± 0.2 GPa at 50 °C up to 9.52 ± 0.2 GPa at 200 °C, then, it went up slightly and reached the maximum value of about 9.56 GPa at 300 °C. This was probably because the improved deposition temperature led to the much stronger ion bombardment effect on the coating surface, which was beneficial to increase the compactness and quality of the coatings [31].

As shown in the experiment results above, it was considered that the deposition temperature had a significant effect on the primary mechanical properties of the coatings. This can be explained in that the higher deposition temperature was beneficial to improve the bombardment energy of sputtering particles, the adsorption and the coagulation in the forming process of the coatings [31], which can effectively increase the coating quality and surface compactness. Thus, as the deposition temperature was increased from 50 °C to 200 °C, the adhesion force, thickness and hardness were significantly enhanced (Figure 6, Figure 8 and Figure 9). However, with the further increasing in temperature, the main coating properties remained basically stable and no longer showed obvious improvement; while the adhesion force decreased, which was due to the thermal stress caused by the much higher temperature.

The thermal stress σ_f_ in the coating can be expressed as follows [31]:σ_f_ = (Δ*α*Δ*TE*_f_)/(1 − *ν*_f_)(1)
where *E*_f_ is the Poisson’s ratio for substrate, *v*_f_ is the Young’s modulus, Δ*α* is the difference of thermal expansion coefficient between the coatings and substrate material and Δ*T* is the difference between the ambient temperature and deposition temperature.

Equation (1) indicates that the thermal stress σ_f_ in the coatings is significantly affected by the deposition temperature. When the deposition temperature was increased from 50 °C to 300 °C, the thermal stress σ_f_ increased obviously according to the formula. This can lead to a decline in the adhesive strength between the coatings and substrate, which was confirmed by the test results of the adhesive strength in Figure 6.

### 3.2. Friction Behaviors

Figure 10 indicates the influence of the deposition temperature on the average friction coefficient of samples at a speed of 250 mm/s and applied load of 10 N. As indicated in this figure, the friction coefficient of the coated samples with various deposition temperatures exhibited a similar upward tendency in the same test conditions. The average values of the friction coefficient for all samples were only about 0.06 to 0.08 at the initial stage of the friction test, and the values gradually increased and stabilized at 0.38 to 0.40 until the coatings were worn out. However, the variation rates of the friction coefficient were different, which can be applied to assess the wear performance of the coated sample. The friction coefficient curve for the T50 sample exhibited the fastest rate of growth, followed by the T120 and T150 ones, and the coefficients of the T180, T200 and T250 ones presented a relatively small rate of increase. Thus, it can be concluded that the samples deposited at temperatures from 180 °C to 250 °C possessed much better lubricating performances and wear resistance.

To evaluate the friction behaviors of the coated samples, SEM and EDX were used to analyze the worn samples. The micrographs of the worn surface and the EDX composition analyses of samples after 90,000 mm sliding distance are shown in Figure 11 and Figure 12. As shown in Figure 11a, large amounts of flaking and delamination of coatings were obviously observed on the wear track of the T50 sample. The composition analyses of points A and B in Figure 11a are shown in Figure 12a,b. It can be found that the coating elements (Mo, S and Zr) were not found in the worn area (Figure 12b), which indicated that the surface coatings of the T50 sample were almost worn out, and large parts of the carbide substrate were exposed. Meanwhile, the Fe element was also identified on the worn area, which was derived from the steel ring in the sliding process. The main wear effects for the T50 sample seemed to be serious flaking and delamination of coatings, and slight adhesive wear, which was consistent with the rapid change of the friction coefficient in Figure 10.

The surface morphology and EDX composition analyses of T120 sample are indicated in Figure 11b and Figure 12c–e. From Figure 11b, obvious coating flaking and abrasions can be found in the worn area, but the wear was mild compared with that of the T50 one. Moreover, a part of the coatings was worn out, the corresponding substrate was already exposed and a certain amount of adhesions of Fe and its oxides were also detected, which were confirmed by the EDX analysis results in Figure 12d,e.

Figure 11c illustrates the micrograph of the worn T150 sample. It can be seen that there were a bit of coating delamination and mechanical scratches on the wear track. Figure 12f,g presents the corresponding EDX analyses of points F and G in Figure 11c. It revealed that the surface adhesions mainly consisted of the coating elements and their oxides. Coating delamination and abrasion were the primary wear effects.

Figure 11d–f indicates that the primary wear effects of the samples at a temperature from 180 °C to 250 °C was just abrasive wear. It was also found that the coatings deposited in this temperature range exhibited excellent wear-resistance in comparison with those prepared at other temperatures. The micrograph of the worn T300 sample (Figure 11g) showed that there existed clear coating flaking, delamination and slight abrasive wear on the wear track, and the wear became heavier compared to the above three samples, which was in line with the rapid increase in friction coefficient in Figure 10. Figure 12h shows the corresponding EDX analysis of point H in Figure 11g. It was evident that there was some adhesive wear, owing to the existence of Fe elements.

As can be seen from the test results, the mechanical properties of the coatings varied, owing to various deposition temperatures that can significantly affect the friction behaviors and wear characteristics. The samples deposited at temperature from 180 °C to 250 °C (T180, T200 and T250 samples) possessed better friction performance and wear resistance, due to the increased adhesion force, hardness, thickness and their combined effects.

## 4. Conclusions

MoS_2_/Zr composite coatings were prepared at different deposition temperatures on the carbide substrate surface, through the MFMS technology combined with the MAIP method. The effects of the deposition temperature on the primary mechanical properties and tribological performances of the composite coatings were investigated. The main conclusions were obtained:As the deposition temperature increased from 50 °C to 300 °C, the coating thickness and micro-hardness gradually increased, while the adhesion force was first increased and then decreased.The friction behaviors for the coated samples varied with the growth of deposition temperatures. The samples within the deposition temperature range from 180 °C to 250 °C possessed excellent friction properties and wear resistance.The main mechanisms responsible for the difference in the friction performance of the coatings were attributed to the changing mechanical performances of coatings caused by temperature variation. The MoS_2_/Zr coatings deposited at the temperature of 180 °C and 200 °C possessed preferable comprehensive mechanical and tribological properties.

## Figures and Tables

**Figure 1 materials-14-05100-f001:**
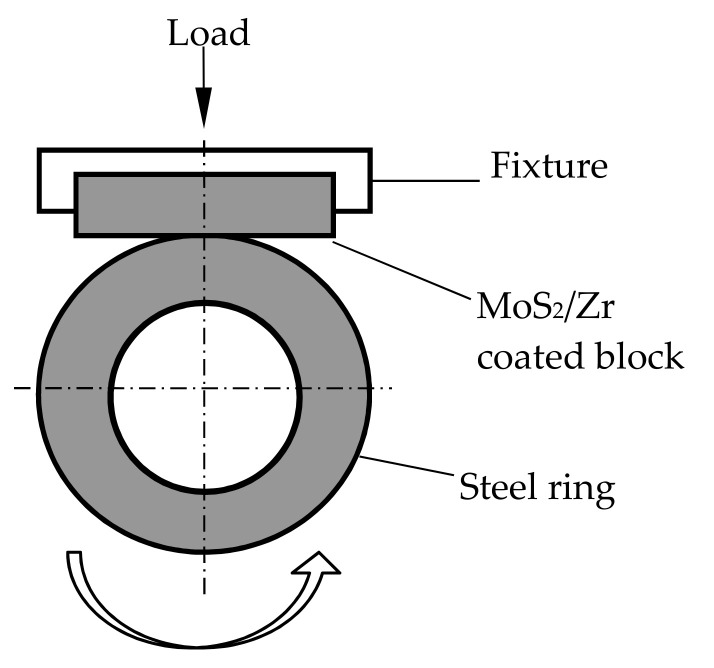
Schematic diagram of block-on-ring friction tester.

**Figure 2 materials-14-05100-f002:**
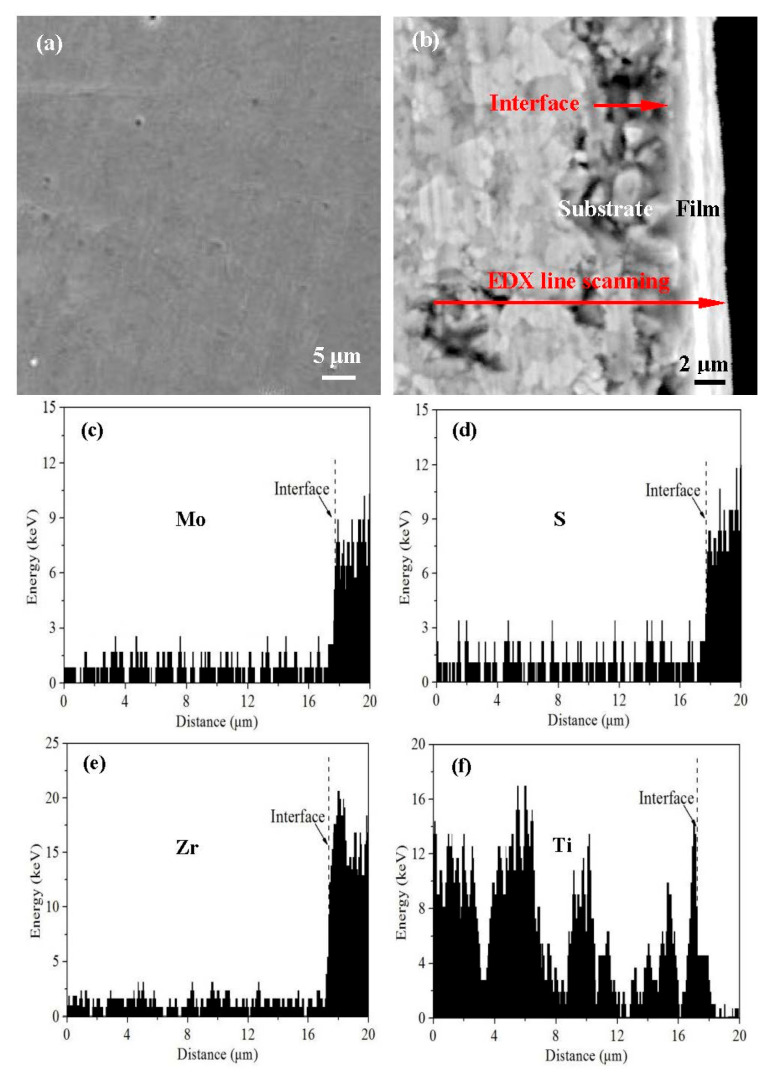
Surface (**a**), cross-section (**b**) morphologies for the T180 sample and the EDX line scanning of Mo (**c**), S (**d**), Zr (**e**) and Ti (**f**) elements in Figure 2b.

**Figure 3 materials-14-05100-f003:**
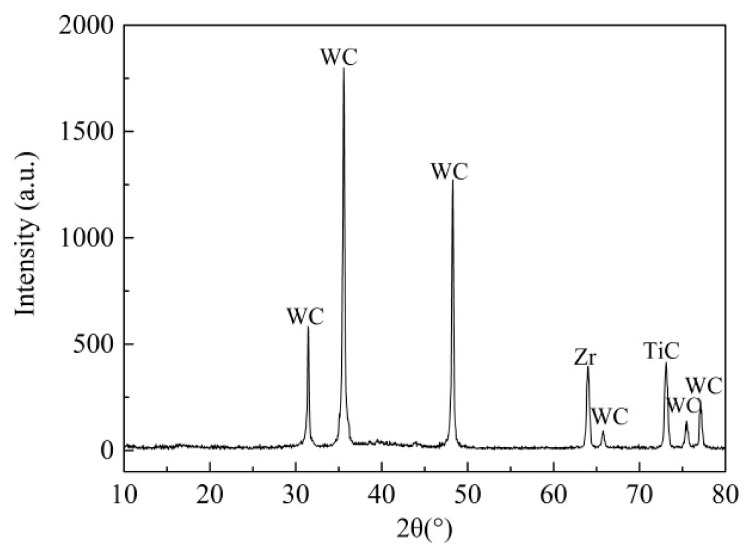
X-ray diffraction analysis of the T180 sample.

**Figure 4 materials-14-05100-f004:**
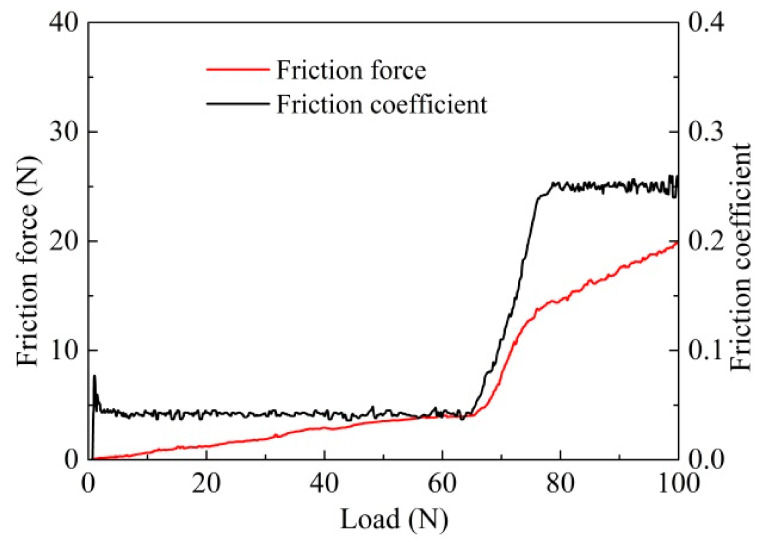
Adhesive strength curve in a scratch test of T180 coated sample.

**Figure 5 materials-14-05100-f005:**
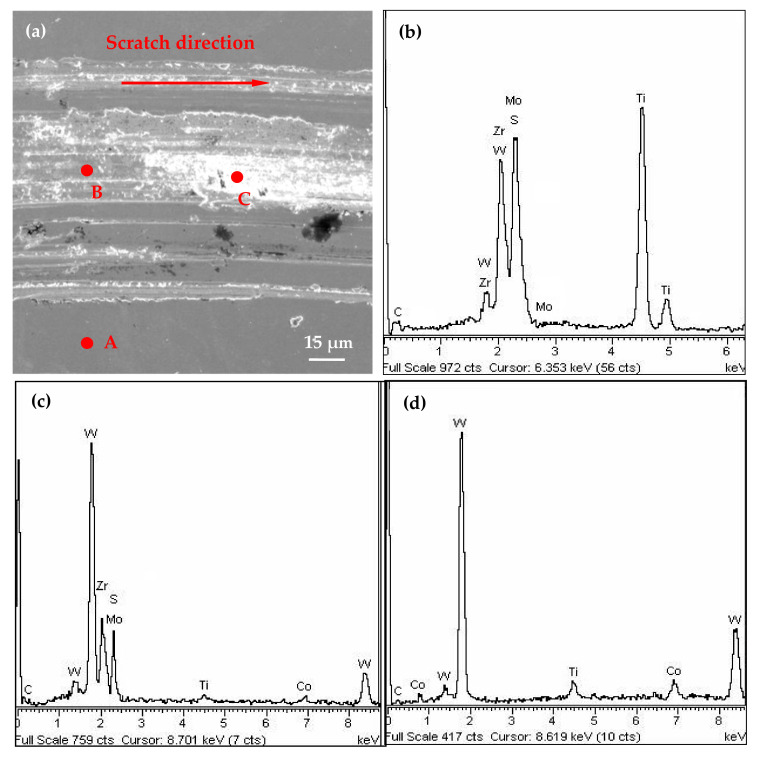
Scratch micrographs (**a**) at the scratch force of 70 N for the T180 sample and corresponding EDX analyses (**b**–**d**) of points A–C in Figure 5a.

**Figure 6 materials-14-05100-f006:**
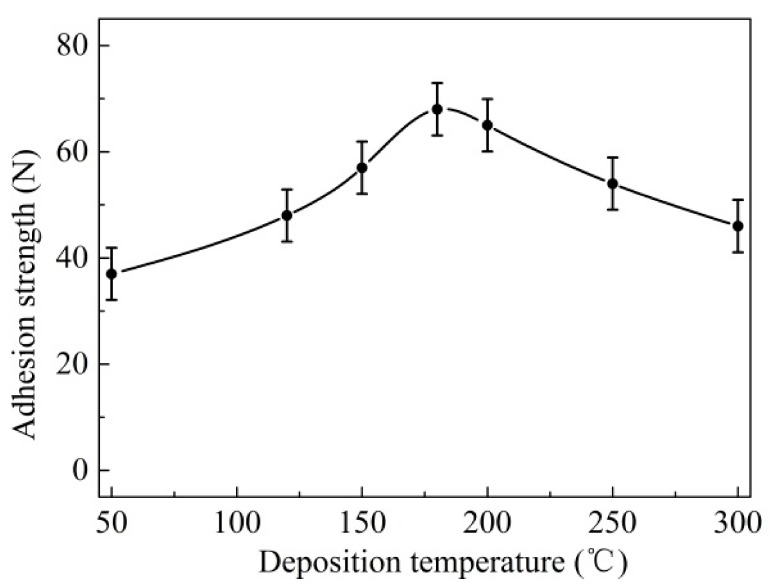
Adhesive strength of coated samples at various deposition temperatures.

**Figure 7 materials-14-05100-f007:**
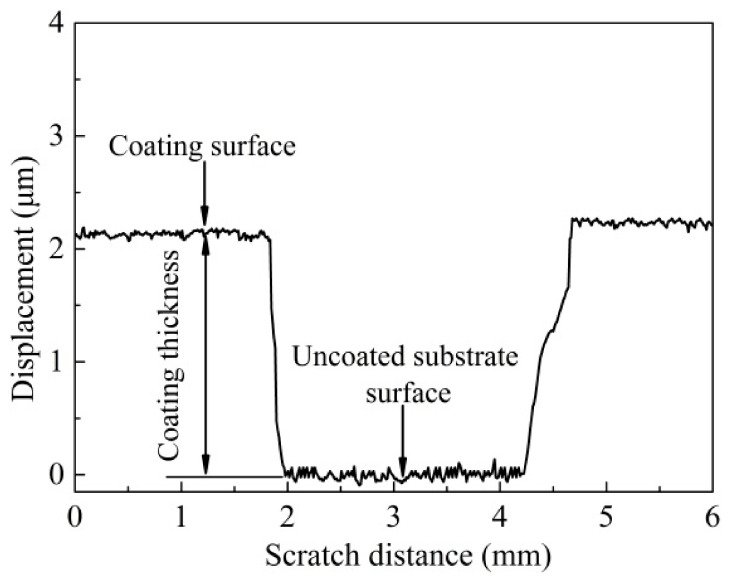
Coating thickness curve in the scratch test of the T180 sample.

**Figure 8 materials-14-05100-f008:**
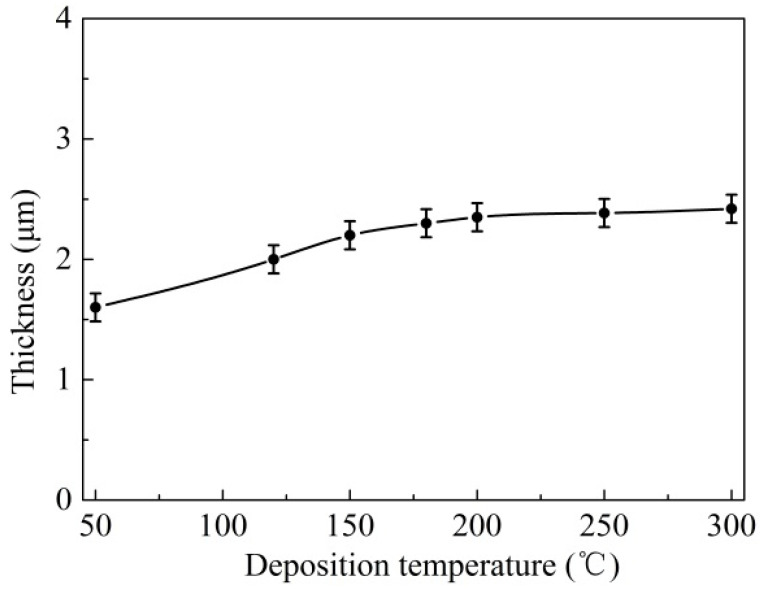
Coating thickness at different deposition temperatures.

**Figure 9 materials-14-05100-f009:**
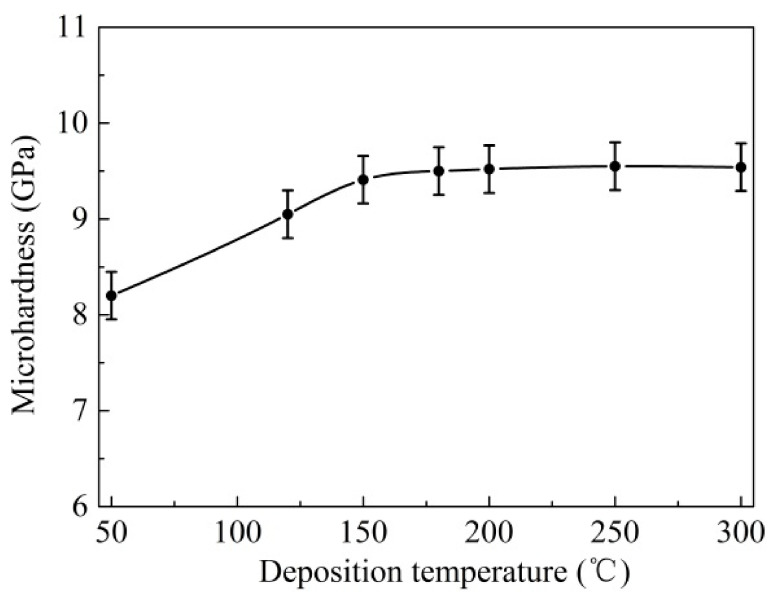
Surface microhardness of coatings at different deposition temperatures.

**Figure 10 materials-14-05100-f010:**
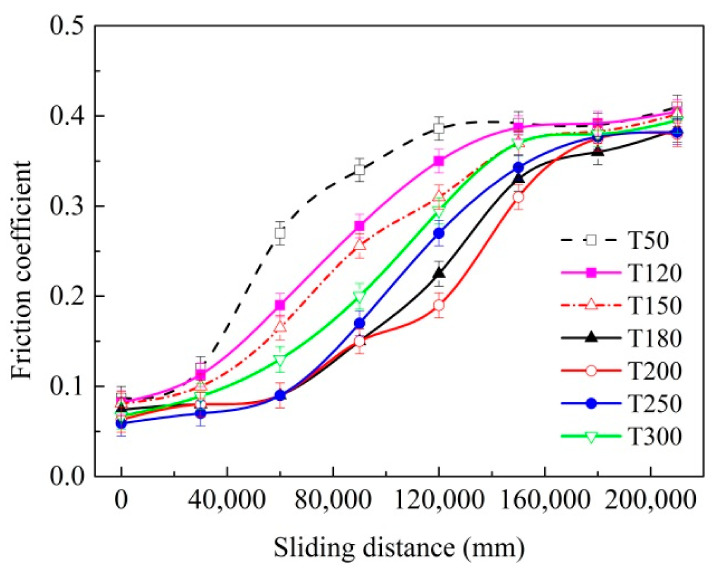
Effect of deposition temperature on the average friction coefficient of coated samples at the sliding speed of 260 mm/s and applied force of 10 N).

**Figure 11 materials-14-05100-f011:**
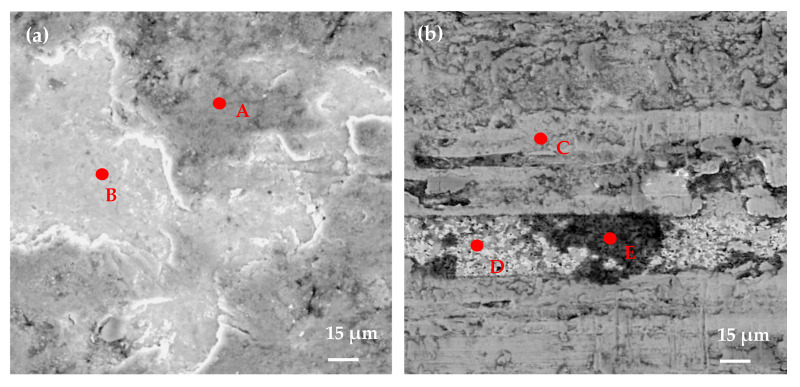
The morphologies of the worn surface of the composite coatings (**a**) T50, (**b**) T120, (**c**) T150, (**d**) T180, (**e**) T200, (**f**) T250, (**g**) T300 (sliding distance 90,000 mm).

**Figure 12 materials-14-05100-f012:**
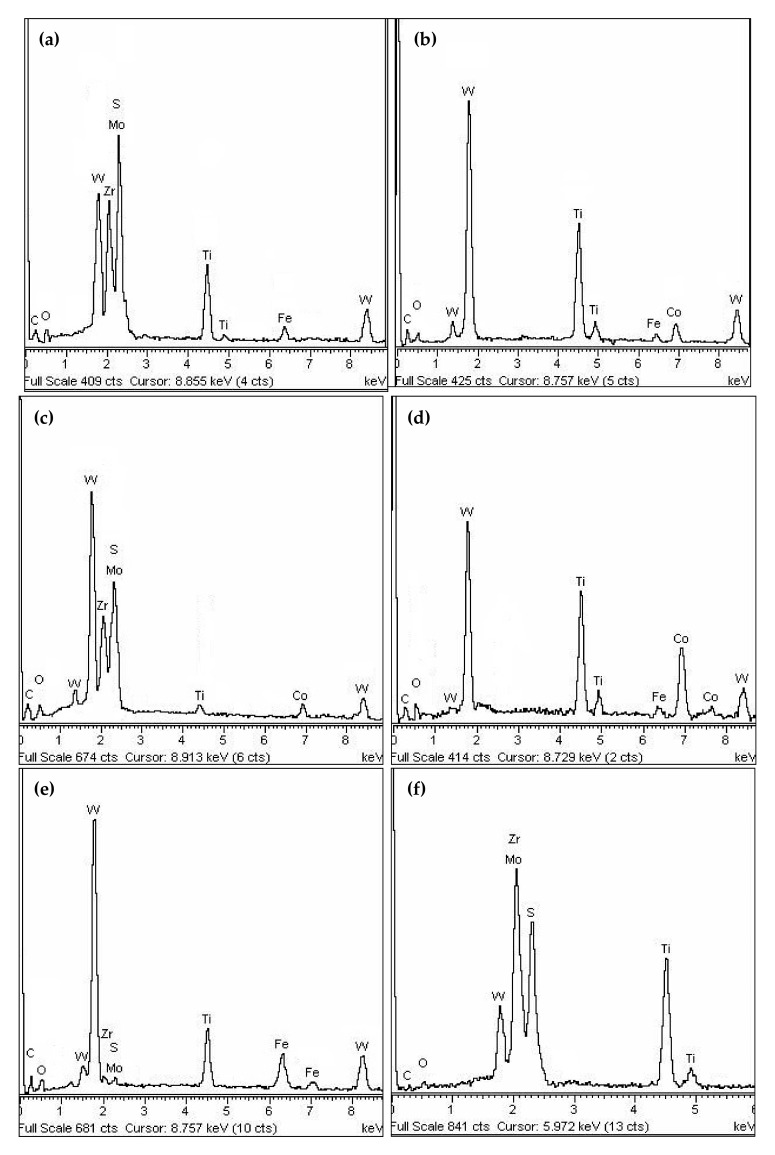
The corresponding composition analysis (**a**–**h**) of points (A)–(H) in Figure 11.

**Table 1 materials-14-05100-t001:** Prepare conditions of MoS_2_-Zr coatings at different deposition temperatures.

Base Pressure (×10^−3^ Pa)	Ar Pressure (Pa)	Bias Voltage (V)	MoS_2_ Current (A)	Zr Current (A)	Deposition Time (min)	Temperature (°C)
6.50	0.50	−200	1.60	60	100	50–300

## Data Availability

The data presented in this study are available on request from the corresponding author.

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
