# Peer review of "Performance of MoS2/Zr Composite Coatings at Different Deposition Temperatures"

_materials, 2021, doi:10.3390/ma14175100_

Round 1
Reviewer 1 Report
The manuscript submitted for review meets the requirements of the periodical. The manuscript contains interesting current research results concerning the mechanical and tribological properties of MoS2/Zr coatings.
Abstract: The object of research, the aim of the work and major conclusions have been given in the abstract. The abstract has been written in accordance with the guidelines from the periodical.
Introduction: The authors explained the need for the research in question while referring to relevant reference literature. Please provide more examples that use coatings to reduce wear. For example, press-fit connections:
- Analysis of the application of ZrN coatings for the mitigation of the development of fretting wear processes at the surfaces of push fit joint elements
- The influence of selected PVD coatings on fretting wear in a clamped joint based on the example of a rail vehicle wheel set
Materials and Methods: The chapter has been prepared in a correct way. The test method has been discussed, the research material described.
Please provide information on the number of samples tested
The test methodology should be supplemented with a graphic representation of the coating application station and a friction test diagram.
Results and discussion: In the chapter containing test results, relevant graphic documentation has been presented and the results have been discussed. In that part of the discussion, the mechanisms permitting the understanding of the test results have been explained.
Conclusion: The manuscript ends with the conclusion providing a summary of the activities undertaken in this work. The chapter is written correctly and exhaustively.
Reference literature: The reference literature consists of 29 items. The authors quote current reference literature relevant to the topic of the article. Quoting was done in accordance with the guidelines from the periodical.
It is suggested that the manuscript should be published in the “Coatings”.
Author Response
Dear Reviewer:
Thank you for your letter and for the reviewers’ comments concerning our manuscript entitled “Performance of MoS2/Zr composite coatings at different deposition temperatures” (ID: materials-1334234). These comments are all valuable and very helpful for revising and improving our paper, as well as the important guiding significance to our researches. We have studied comments carefully and have made corrections which we hope to meet with approval. In order to clearly show the changes of the manuscript to the editors and reviewers, the function of "Track Changes" in Microsoft Word was used, and the revised manuscript can be displayed properly by clicking the final status without markup in the review tab. The main corrections in the paper and the responds to the reviewer’s comments are as flowing:
Responds to the reviewer’s comments:
(x) I would not like to sign my review report
( ) I would like to sign my review report
English language and style
( ) Extensive editing of English language and style required
( ) Moderate English changes required
(x) English language and style are fine/minor spell check required
( ) I don't feel qualified to judge about the English language and style
|
Yes |
Can be improved |
Must be improved |
Not applicable |
|
|
Does the introduction provide sufficient background and include all relevant references? |
( ) |
(x) |
( ) |
( ) |
|
Is the research design appropriate? |
(x) |
( ) |
( ) |
( ) |
|
Are the methods adequately described? |
( ) |
(x) |
( ) |
( ) |
|
Are the results clearly presented? |
(x) |
( ) |
( ) |
( ) |
|
Are the conclusions supported by the results? |
(x) |
( ) |
( ) |
( ) |
Comments and Suggestions for Authors
The manuscript submitted for review meets the requirements of the periodical. The manuscript contains interesting current research results concerning the mechanical and tribological properties of MoS2/Zr coatings.
Abstract: The object of research, the aim of the work and major conclusions have been given in the abstract. The abstract has been written in accordance with the guidelines from the periodical.
Introduction: The authors explained the need for the research in question while referring to relevant reference literature. Please provide more examples that use coatings to reduce wear. For example, press-fit connections:
Analysis of the application of ZrN coatings for the mitigation of the development of fretting wear processes at the surfaces of push fit joint elements.
The influence of selected PVD coatings on fretting wear in a clamped joint based on the example of a rail vehicle wheel set.
Response:
Considering the Reviewer’s suggestion, two relevant literatures that use coatings to reduce wear are added in the Reference part.
[4] Kowalski, S. The influence of selected PVD coatings on fretting wear in a clamped joint based on the example of a rail vehicle wheel set. Eksploat. Niezawodn. 2017, 20(1), 1-8.
[5] Kowalski, S.; Cygnar, M.; Cieślikowski, B. Analysis of the application of ZrN coatings for the mitigation of the development of fretting wear processes at the surfaces of push fit joint elements. P. I. Mech. E. J-J. Eng. 2020, 234 (8), 1208-1221.
Materials and Methods: The chapter has been prepared in a correct way. The test method has been discussed, the research material described.
Please provide information on the number of samples tested.
Response:
In Line 91 of section 2.1, all the tests were performed three times and the average values were taken. Then, there were three samples for each test.
The test methodology should be supplemented with a graphic representation of the coating application station and a friction test diagram.
Response:
Considering the Reviewer’s suggestion, the friction test diagram was supplemented in Figure 1.
Results and discussion: In the chapter containing test results, relevant graphic documentation has been presented and the results have been discussed. In that part of the discussion, the mechanisms permitting the understanding of the test results have been explained.
Conclusion: The manuscript ends with the conclusion providing a summary of the activities undertaken in this work. The chapter is written correctly and exhaustively.
Reference literature: The reference literature consists of 29 items. The authors quote current reference literature relevant to the topic of the article. Quoting was done in accordance with the guidelines from the periodical.
It is suggested that the manuscript should be published in the “Coatings”.
We tried our best to improve the manuscript and made some changes in the manuscript. These changes will not influence the content and framework of the paper. And here we did not list the changes but can be tracked down in revised paper by clicking "Track Changes".
We appreciate for your warm work earnestly, and hope that the correction will meet with approval.
Once again, thank you very much for your comments and suggestions.
Best Regards,
Corresponding author: Wenlong Song
E-mail: wlsong@jnxy.sdu.edu.cn
Reviewer 2 Report
The paper is a study on the effects of deposition temperature on the mechanical wear and tribological properties of MoS2/Zr coatings, deposited on a carbide surface via vacuum deposition. The authors show that increasing the deposition temperature 50 - 300C leads to several effects on the as-deposited coatings - the least of them is that the rate of deposition increases (or possibly the density of the deposited coating), and more pronounced effects are the increase in micro hardness in the range 50-150C, with no further increase at higher temperatures, as well as that the adhesion strength and lubricative properties are modulated with temperature with an optimal properties for samples deposited 150-200C. The paper is application oriented and can be a good guideline for the development of such coatings, which are important in the design of cutting tools, e.g. The paper has a clear structure, and is with a relatively good English presentation, however, still there are some expressions that can be improved, here are a few examples:
Line 33 - the list of materials in the brackets, “etc.” could be removed and an “e.g.” can be added in the beginning.
Line 35 - “more excellent” can simply be “with excellent”.
Lines 39-40 - the sentence “The mechanism…” should be in present tense.
Lines 53-59 - should probably be in present tense.
Lines 59 - “The prepare conditions” should be replaced.
Lines 69-70 there is a list of samples, deposited at different temperatures. Maybe these brackets and listed temperature values can be omitted. Obviously, the naming convention is TXXX (where XXX is the deposition temperature in degrees Celsius). Alternatively, you can simply list the sample names first, and then follow with the temperatures (this is the case of proper use of “, respectively.”).
On line 75, it states that “a 100N/min scratch speed”. I am not sure if a different unit should be here ? Or the this is the rate of load increase ? Please clarify.
Line 87 “ultrasonically rinsed with acetone”. It is a strange expression, I guess that you mean that it was ultrasonically cleaned in acetone solvent, and then rinsed (it is not really important in this case, but dubious statements always attract attention to one when reading).
Line 96. There is a “3.1. Subsection”, which I guess is a leftover from the template.
Line 99. “without the columnar crystals”. Now. I understand that probably here what is meant is that no columnar growth, which is typical for sputtered coatings, is observed. I think that is should be corrected (also, because these are no “crystals”, but usually polycrystalline columns, in your case even amorphous, it seems).
Line 106. “one mutant of Ti content” I hope that this is an Autocorrect error. Please fix.
Line 111. “by adding a proper amount of Zr element can cause the vanishing of pure MoS2..”. This should be rewritten. What is a “proper amount”. And the term “vanishing of MoS2 phase” I saw used in another paper, cited in the list. Lines 112-115 should also be rewritten. The statement there is not revealed by the “test results”, and is based on literature, as I can see. It could be true, if there was a reference sample, based only on MoS2 (and no Zr), so that it can be demonstrated that the addition of Zr destabilise it (an lattice parameter seems improper for an amorphous phase).
Figure 2. One of the peaks (around 64 2Theta degrees) is marked as Zr. Can the authors confirm (and explain in the text) what the crystalline phases in the diffractogram are ? If metallic Zr is implied there, there should also be reflection peaks at lower angles (possibly overlapping with the WC ones). In all cases this figure should be improved and some more information about the XRD analysis should be provide (i.e. where were the reference positions obtained from). If XRD information is available for the other temperatures it also can be provided.
Figure 4. It could be a bit confusing that points A, B and C are on figures (b), (c), (d). Fortunately on Fig 10 and 11 it is easier to follow
Line 190. “Formula (1)” - better to use Equation (1).
Line 275. “relatively excellent friction properties” should be corrected.
Author Response
Dear Reviewer:
Thank you for your letter and for the reviewers’ comments concerning our manuscript entitled “Performance of MoS2/Zr composite coatings at different deposition temperatures” (ID: materials-1334234). These comments are all valuable and very helpful for revising and improving our paper, as well as the important guiding significance to our researches. We have studied comments carefully and have made corrections which we hope to meet with approval. In order to clearly show the changes of the manuscript to the editors and reviewers, the function of "Track Changes" in Microsoft Word was used, and the revised manuscript can be displayed properly by clicking the final status without markup in the review tab. The main corrections in the paper and the responds to the reviewer’s comments are as flowing:
Responds to the reviewer’s comments:
(x) I would not like to sign my review report
( ) I would like to sign my review report
English language and style
( ) Extensive editing of English language and style required
( ) Moderate English changes required
(x) English language and style are fine/minor spell check required
( ) I don't feel qualified to judge about the English language and style
|
Yes |
Can be improved |
Must be improved |
Not applicable |
|
|
Does the introduction provide sufficient background and include all relevant references? |
( ) |
(x) |
( ) |
( ) |
|
Is the research design appropriate? |
(x) |
( ) |
( ) |
( ) |
|
Are the methods adequately described? |
(x) |
( ) |
( ) |
( ) |
|
Are the results clearly presented? |
( ) |
(x) |
( ) |
( ) |
|
Are the conclusions supported by the results? |
(x) |
( ) |
( ) |
( ) |
Comments and Suggestions for Authors
The paper is a study on the effects of deposition temperature on the mechanical wear and tribological properties of MoS2/Zr coatings, deposited on a carbide surface via vacuum deposition. The authors show that increasing the deposition temperature 50 - 300C leads to several effects on the as-deposited coatings - the least of them is that the rate of deposition increases (or possibly the density of the deposited coating), and more pronounced effects are the increase in micro hardness in the range 50-150C, with no further increase at higher temperatures, as well as that the adhesion strength and lubricative properties are modulated with temperature with an optimal properties for samples deposited 150-200C. The paper is application oriented and can be a good guideline for the development of such coatings, which are important in the design of cutting tools, e.g. The paper has a clear structure, and is with a relatively good English presentation, however, still there are some expressions that can be improved, here are a few examples:
Line 33 - the list of materials in the brackets, “etc.” could be removed and an “e.g.” can be added in the beginning.
Response:
Considering the Reviewer’s suggestion, the “etc.” was removed and an “e.g.” was added in the beginning.
Line 35 - “more excellent” can simply be “with excellent”.
Response:
Considering the Reviewer’s suggestion, the term “more excellent” was replaced by “with excellent”.
Lines 39-40 - the sentence “The mechanism…” should be in present tense.
Response:
Considering the Reviewer’s suggestion, the term “The mechanism…was” was changed to “The mechanism…is”.
Lines 53-59 - should probably be in present tense.
Response:
Considering the Reviewer’s suggestion, the sentence in Lines 53-59 was changed to be in present tense.
Lines 59 - “The prepare conditions” should be replaced.
Response:
Considering the Reviewer’s suggestion, the term “The prepare conditions” was changed to “The deposition conditions”.
Lines 69-70 there is a list of samples, deposited at different temperatures. Maybe these brackets and listed temperature values can be omitted. Obviously, the naming convention is TXXX (where XXX is the deposition temperature in degrees Celsius). Alternatively, you can simply list the sample names first, and then follow with the temperatures (this is the case of proper use of “, respectively.”).
Response:
Considering the Reviewer’s suggestion, the statement about the samples deposited at different temperatures was revised as below:
The MoS2/Zr coatings deposited at different temperatures were named T50, T120, T150, T180, T200, T250 and T300, according to the deposition temperatures.
On line 75, it states that “a 100N/min scratch speed”. I am not sure if a different unit should be here ? Or the this is the rate of load increase ? Please clarify.
Response:
We are very sorry for our negligence, the term “a 100N/min scratch speed” was changed to “a 100 N/min load increasing rate”.
Line 87 “ultrasonically rinsed with acetone”. It is a strange expression, I guess that you mean that it was ultrasonically cleaned in acetone solvent, and then rinsed (it is not really important in this case, but dubious statements always attract attention to one when reading).
Response:
Considering the Reviewer’s suggestion, the term “ultrasonically rinsed with acetone” was changed to “cleaned ultrasonically with acetone”.
Line 96. There is a “3.1. Subsection”, which I guess is a leftover from the template.
Response:
Considering the Reviewer’s suggestion, the term “3.1. Subsection” was deleted.
Line 99. “without the columnar crystals”. Now. I understand that probably here what is meant is that no columnar growth, which is typical for sputtered coatings, is observed. I think that is should be corrected (also, because these are no “crystals”, but usually polycrystalline columns, in your case even amorphous, it seems).
Response:
Considering the Reviewer’s suggestion, the term “without the columnar crystals” was deleted.
Line 106. “one mutant of Ti content” I hope that this is an Autocorrect error. Please fix.
Response:
Considering the Reviewer’s suggestion, the term “one mutant of Ti content” was changed to “one peak of Ti content”.
Line 111. “by adding a proper amount of Zr element can cause the vanishing of pure MoS2.”. This should be rewritten. What is a “proper amount”. And the term “vanishing of MoS2 phase” I saw used in another paper, cited in the list. Lines 112-115 should also be rewritten. The statement there is not revealed by the “test results”, and is based on literature, as I can see. It could be true, if there was a reference sample, based only on MoS2 (and no Zr), so that it can be demonstrated that the addition of Zr destabilise it (an lattice parameter seems improper for an amorphous phase).
Response:
Considering the Reviewer’s suggestion, the statement “It was explained that the MoS2/Zr coatings by adding a proper amount of Zr element can cause the vanishing of pure MoS2 crystalline phase” was corrected as “It was explained that the addition of Zr element could destabilise it”.
Figure 2. One of the peaks (around 64 2Theta degrees) is marked as Zr. Can the authors confirm (and explain in the text) what the crystalline phases in the diffractogram are ? If metallic Zr is implied there, there should also be reflection peaks at lower angles (possibly overlapping with the WC ones). In all cases this figure should be improved and some more information about the XRD analysis should be provide (i.e. where were the reference positions obtained from). If XRD information is available for the other temperatures it also can be provided.
Response:
Figure 2 shows the X-ray diffraction analysis result of the MoS2/Zr coating of the T180 sample. It revealed that a very broad band pattern indicating a structure consisting of quasi-amorphous MoS2, and so it would appear that the addition of Zr into MoS2 coating led to the vanishing of MoS2 crystalline phase. It was also reported by the previous findings [19-21,24,28] that the MoS2/metal composite coatings by adding of metal (Ti, Cr, Zr, etc.) to MoS2 to form has resulted in the distortion of MoS2 lattice parameters and X-ray amorphous microstructure. The XRD patterns of coated samples with different temperatures are almost similar owing to the vanishing of MoS2, and we won't go into details here.
Considering the Reviewer’s suggestion, the position information about the XRD analysis was provide in the 2nd paragraph of Section 3.1
Figure 4. It could be a bit confusing that points A, B and C are on figures (b), (c), (d). Fortunately on Fig 10 and 11 it is easier to follow
Response:
Figure. 4 shows the scratch micrographs and the corresponding EDX analysis results of points A, B and C. It can be seen that there existed serious mechanical plows and delamination in the scratch area (Figure. 4(a)). Figure. 4(b) shows the EDX analysis of point A on the coating surface without scratch, and there existed large amount of coating elements. Figures. 4(c) and (d) show the EDX analysis results of points in the scratch track. Figure. 4(c) is the analysis result of point B, Where the scratch was relatively light. It can be shown that the carbide substrate was partly exposed and a small amount of coating elements could be detected because of the serious scratch. Figure. 4(d) is the analysis result of point C, which suffered more severe scratches. It indicated that there existed no coating element due to the high scratch load, which was consistent with the high values of friction force and friction coefficient in Figure. 3.
Therefore, the process of scratch test is the process of gradual peeling of the coating, and the friction force and friction coefficient gradually rise due to the wear and tear of coatings during the scratch process.
Line 190. “Formula (1)” - better to use Equation (1).
Response:
Considering the Reviewer’s suggestion, “Formula (1)” was changed to “Equation (1)”.
Line 275. “relatively excellent friction properties” should be corrected.
Response:
Considering the Reviewer’s suggestion, the term “relatively excellent friction properties” was changed to “excellent friction properties”.
We tried our best to improve the manuscript and made some changes in the manuscript. These changes will not influence the content and framework of the paper. And here we did not list the changes but can be tracked down in revised paper by clicking "Track Changes".
We appreciate for your warm work earnestly, and hope that the correction will meet with approval.
Once again, thank you very much for your comments and suggestions.
Best Regards,
Corresponding author: Wenlong Song
E-mail: wlsong@jnxy.sdu.edu.cn
Reviewer 3 Report
The article „Performance of MoS2/Zr composite coatings at different deposition temperatures“ by W. Song et al. is a clear and well-presented description of the mechanical and tribological properties of MoS2/Zr coatings of different composition deposited onto a carbide substrate a different temperatures. The methods use are appropriate, the experimental conditions are clearly described, and the conclusions are supported by the obtained data and convincing.
In principle the paper can be published as it stands. However, a few linguistic corrections would improve the readability of the work:
Page 2, line 68: “The prepare conditions…” better “The preparation conditions…”
Page 3, line 112: “the test results were also revealed …” better “the test results also revealed that….”
Page 4, line 128: “initial stag….” Better “…initial stage …”
Page 6, line 162: What do you mean by “deposition power”? Do you mean “sticking coefficient”?
And if you really mean “sticking coefficient” then you should explain why the
“sticking coefficient” increases with increasing temperature.
Page 7, line 191: “…temperature was improved…” better “…. Temperature was increased…”
Page 7, line 212: “The worn micrographs….” Better “The micrographs of the worn surface…” (The micrographs are not worn; the surface is worn.)
Page 7, line 221: “Thus, the main wear forms…” better “The main wear effects…”
Page 8, line 237: “… the primary wear types… better “…. the primary wear effects…”
Pge 10, line 248: “The worn micrograph of the T300 sample…” better “The micrograph of the worn T300 sample…” (see above)
Page 10, line 258-259: Better: “As can be seen from the test results, the mechanical properties of the coatings varied owing…”
Finally: Fig.1b would be clearer if you would mark the “interface” (between substrate and film) as well as the film surface by a dashed (for instance red) line.
Author Response
Dear Reviewer:
Thank you for your letter and for the reviewers’ comments concerning our manuscript entitled “Performance of MoS2/Zr composite coatings at different deposition temperatures” (ID: materials-1334234). These comments are all valuable and very helpful for revising and improving our paper, as well as the important guiding significance to our researches. We have studied comments carefully and have made corrections which we hope to meet with approval. In order to clearly show the changes of the manuscript to the editors and reviewers, the function of "Track Changes" in Microsoft Word was used, and the revised manuscript can be displayed properly by clicking the final status without markup in the review tab. The main corrections in the paper and the responds to the reviewer’s comments are as flowing:
Responds to the reviewer’s comments:
(x) I would not like to sign my review report
( ) I would like to sign my review report
English language and style
( ) Extensive editing of English language and style required
(x) Moderate English changes required
( ) English language and style are fine/minor spell check required
( ) I don't feel qualified to judge about the English language and style
Yes |
Can be improved |
Must be improved |
Not applicable |
|
|
Does the introduction provide sufficient background and include all relevant references? |
(x) |
( ) |
( ) |
( ) |
|
Is the research design appropriate? |
(x) |
( ) |
( ) |
( ) |
|
Are the methods adequately described? |
(x) |
( ) |
( ) |
( ) |
|
Are the results clearly presented? |
(x) |
( ) |
( ) |
( ) |
|
Are the conclusions supported by the results? |
(x) |
( ) |
( ) |
( ) |
The article “Performance of MoS2/Zr composite coatings at different deposition temperatures” by W. Song et al. is a clear and well-presented description of the mechanical and tribological properties of MoS2/Zr coatings of different composition deposited onto a carbide substrate a different temperatures. The methods use are appropriate, the experimental conditions are clearly described, and the conclusions are supported by the obtained data and convincing.
In principle the paper can be published as it stands. However, a few linguistic corrections would improve the readability of the work:
Page 2, line 68: “The prepare conditions…” better “The preparation conditions…”
Response:
Considering the Reviewer’s suggestion, “The prepare conditions…” was changed to “The preparation conditions…”.
Page 3, line 112: “the test results were also revealed …” better “the test results also revealed that….”
Response:
Considering the Reviewer’s suggestion, “the test results were also revealed …” was changed to “the test results also revealed that….”.
Page 4, line 128: “initial stag….” Better “…initial stage …”
Response:
Considering the Reviewer’s suggestion, “initial stag….” was changed to “…initial stage …”.
Page 6, line 162: What do you mean by “deposition power”? Do you mean “sticking coefficient”? And if you really mean “sticking coefficient” then you should explain why the “sticking coefficient” increases with increasing temperature.
Response:
Considering the Reviewer’s suggestion, “deposition power” was changed to “sputtering energy”.
Page 7, line 191: “…temperature was improved…” better “…. Temperature was increased…”
Response:
Considering the Reviewer’s suggestion, “…temperature was improved…” was changed to “…. Temperature was increased…”.
Page 7, line 212: “The worn micrographs….” Better “The micrographs of the worn surface…” (The micrographs are not worn; the surface is worn.)
Response:
Considering the Reviewer’s suggestion, “The worn micrographs….” was changed to “The micrographs of the worn surface…”.
Page 7, line 221: “Thus, the main wear forms…” better “The main wear effects…”
Response:
Considering the Reviewer’s suggestion, “Thus, the main wear forms…” was changed to “The main wear effects…”.
Page 8, line 237: “… the primary wear types… better “…. the primary wear effects…”
Response:
Considering the Reviewer’s suggestion, “… the primary wear types…” was changed to “…. the primary wear effects…”.
Page 10, line 248: “The worn micrograph of the T300 sample…” better “The micrograph of the worn T300 sample…” (see above)
Response:
Considering the Reviewer’s suggestion, “The worn micrograph of the T300 sample…” was changed to “The micrograph of the worn T300 sample… ”.
Page 10, line 258-259: Better: “As can be seen from the test results, the mechanical properties of the coatings varied owing…”
Response:
Considering the Reviewer’s suggestion, “…it was believed that the varied mechanical properties of the coatings owing…” was changed to “the mechanical properties of the coatings varied owing…”.
Finally: Fig.1b would be clearer if you would mark the “interface” (between substrate and film) as well as the film surface by a dashed (for instance red) line.
Response:
Considering the Reviewer’s suggestion, the interface between the substrate and film was marked by a dashed line.
We tried our best to improve the manuscript and made some changes in the manuscript. These changes will not influence the content and framework of the paper. And here we did not list the changes but can be tracked down in revised paper by clicking "Track Changes".
We appreciate for your warm work earnestly, and hope that the correction will meet with approval.
Once again, thank you very much for your comments and suggestions.
Best Regards,
Corresponding author: Wenlong Song
E-mail: wlsong@jnxy.sdu.edu.cn
Round 2
Reviewer 2 Report
The manuscript has seen some improvements and is now suitable for publication in MDPI Materials in its current form. I would like to thank the authors for considering and accepting reviewers' suggestions and congratulate them on their successful work. Still, please do a careful read-through and spell-check for any residual typos and mistake during the proofreading phase and prior publication.